# High Serum Ferritin Levels Are Associated with Sarcopenia in Patients Undergoing Chronic Hemodialysis

**DOI:** 10.3390/nu17142323

**Published:** 2025-07-15

**Authors:** Mayuko Hori, Hiroshi Takahashi, Chika Kondo, Asami Takeda, Kunio Morozumi, Shoichi Maruyama

**Affiliations:** 1Department of Nephrology, Masuko Memorial Hospital, Nakamura-ku, Nagoya 453-8566, Aichi, Japan; 2Department of Nephrology, Fujita Health University School of Medicine, Toyoake 470-1192, Aichi, Japan; 3Department of Nephrology, Nagoya University Graduate School of Medicine, Showa-ku, Nagoya 466-8550, Aichi, Japan

**Keywords:** ferritin, hemodialysis, iron, sarcopenia, skeletal muscle

## Abstract

**Background/Objectives:** Patients undergoing hemodialysis frequently receive oral or intravenous iron supplementation to treat iron-deficiency anemia and enhance the efficacy of erythropoiesis-stimulating agents. However, this approach may lead to iron overload. Experimental studies have suggested that iron overload may contribute to the development of sarcopenia through oxidative stress and inflammation. This study aimed to investigate the association between iron status and sarcopenia in patients undergoing hemodialysis. **Methods:** Serum ferritin levels were measured, and sarcopenia was assessed using the Asian Working Group for Sarcopenia criteria in 104 stable outpatients undergoing maintenance hemodialysis therapy. **Results:** Sarcopenia was identified in 25 (24.0%) patients. Serum ferritin levels were significantly higher in patients with sarcopenia than in those without (median: 170.6 ng/mL vs. 92 ng/mL, *p* = 0.023). An increase of 10 ng/mL in serum ferritin levels was independently associated with sarcopenia. The high-ferritin group (≥132 ng/mL as a cutoff value determined using receiver operating characteristic curve analysis) exhibited a higher prevalence of sarcopenia compared with the low-ferritin group (37.3% vs. 11.3%, *p* = 0.001). Furthermore, serum ferritin levels were negatively correlated with skeletal muscle mass and skeletal muscle strength, which constitute the components of the sarcopenia diagnostic criteria. **Conclusions:** Elevated serum ferritin levels were independently associated with sarcopenia in patients undergoing hemodialysis. This finding implies that excessive iron supplementation may contribute to the progression of sarcopenia. Routine evaluation of iron status and careful assessment of the necessity for iron therapy are recommended in this population.

## 1. Introduction

Iron is an essential trace metal required by nearly all living organisms, playing a critical role in various metabolic processes [1]. Ferritin, an iron-storage protein, is regulated post-transcriptionally according to cellular iron status, with low serum ferritin levels indicating iron depletion [2]. Iron deficiency is commonly observed in patients undergoing hemodialysis (HD) due to decreased gastrointestinal absorption and blood loss associated with uremia-induced platelet dysfunction, the use of hemodialyzers, frequent blood sampling, and the use of vascular access sites [3]. To correct iron-deficiency anemia (IDA) and enhance the effectiveness of erythropoiesis-stimulating agents (ESAs), oral or intravenous iron supplementation is frequently administered, which may result in iron overload [4]. Iron in the skeletal muscle is also essential for various metabolic functions; hence, excess iron can promote oxidative stress and trigger inflammatory responses. Recent studies have shown that iron overload in skeletal muscles may contribute to the development of sarcopenia [5]. However, whether iron status is associated with sarcopenia in patients undergoing HD remains unclear. Sarcopenia is a progressive and generalized skeletal muscle disorder characterized by a decline in muscle mass, strength, and physical performance [6,7]. Sarcopenia is highly prevalent in patients with end-stage renal disease (ESRD) undergoing dialysis (13.7–40.0%) and is associated with increased mortality [8,9,10,11]. Given these concerns, a better understanding of the underlying mechanisms of sarcopenia in patients undergoing HD is warranted. This study aimed to investigate the association between iron status and sarcopenia in patients undergoing HD.

## 2. Materials and Methods

### 2.1. Study Population

This cross-sectional study was conducted at Masuko Memorial Hospital. A total of 104 outpatients who had been undergoing HD for at least 3 months and whose serum ferritin levels and sarcopenia status were evaluated within 2 months between October 2021 and January 2023 were enrolled in the study. Patients with acute illnesses or injuries requiring hospitalization were excluded from the study. The study was conducted in accordance with the principles of the Declaration of Helsinki and approved by the hospital ethics committee (ethics approval number: MR6-9). The requirement for written informed consent was waived owing to the retrospective nature of the study, which utilized data obtained for clinical purposes. Instead, information regarding the option to opt out was made available on the hospital’s website.

### 2.2. Covariates

Data on the following covariates were collected: demographics (age, sex, body mass index [BMI], dialysis vintage, comorbidities, primary cause of kidney disease, and use of iron supplementation and an ESA) and laboratory measurements (serum ferritin levels, serum iron levels, total iron-binding capacity levels, transferrin saturation levels, C-reactive protein [CRP] levels, serum albumin [Alb] levels, hemoglobin [Hb] levels, and intact parathyroid hormone levels). Blood samples were collected from the arteriovenous (AV) fistula or AV graft immediately before the first weekly HD session. Comorbidities were evaluated using a comorbidity index developed for patients undergoing dialysis. This index includes the primary causes of ESRD, atherosclerotic heart disease, congestive heart failure, cerebrovascular accident/transient ischemic attack, peripheral vascular disease, dysrhythmia, other cardiac conditions, chronic obstructive pulmonary disease, gastrointestinal bleeding, liver disease, cancer, and diabetes [12].

### 2.3. Assessment of Skeletal Muscle Mass

Skeletal muscle mass was evaluated via bioelectrical impedance analysis using the InBody 430 device (In Body Japan Co., Ltd., Tokyo, Japan). The skeletal muscle mass index (SMI) was calculated using the following formula: SMI (kg/m^2^) = appendicular skeletal muscle mass (kg)/height squared (m^2^). Low muscle mass was defined based on the Asian Working Group for Sarcopenia (AWGS) criteria, with SMI thresholds of <7.0 kg/m^2^ for men and <5.7 kg/m^2^ for women [13].

### 2.4. Assessment of Skeletal Muscle Strength

Skeletal muscle strength was evaluated by measuring handgrip strength using a digital dynamometer (TKK 5101 Grip-D, Takei, Tokyo, Japan). The maximal isometric voluntary contractions of both hands were measured twice in an upright position, and the highest value was used for analysis. According to the AWGS criteria, low muscle strength was defined as handgrip strength values of <28 kg for men and <18 kg for women [13].

### 2.5. Assessment of the Physical Performance

Physical performance was assessed using the Short Physical Performance Battery (SPPB), which includes evaluations of usual gait speed, repeated chair stands, and standing balance, as described in established protocols [14]. The total SPPB score ranges from 0 to 12, with each component scored on a scale of 0–4 points. Low physical performance was defined as an SPPB score ≤ 9 for both sexes.

### 2.6. Definition of Sarcopenia

Sarcopenia was defined according to the AWGS 2019 criteria [13]. A diagnosis was made when low muscle mass was present in combination with either low muscle strength and/or low physical performance.

### 2.7. Statistical Analysis

The patients’ characteristics are expressed as the means (standard deviations) or medians (interquartile ranges) for continuous variables and as percentages for categorical variables. The differences between the two groups were evaluated using Student’s *t*-test or the Mann–Whitney U test for continuous variables and the chi-square test for categorical variables. The associations between continuous variables were examined using Spearman’s correlation coefficients. The predictive value of serum ferritin levels for sarcopenia was examined from the same dataset used for the outcomes analysis using receiver operating characteristic (ROC) curve analysis, and the optimal cutoff point was determined based on the highest Youden index (sensitivity + specificity − 1) [15].

Odds ratios (ORs) and 95% confidence intervals (CIs) were calculated to assess the associations between covariates and sarcopenia. The OR was reported as the exponentiated beta coefficient from the logistic regression model. To identify the independent predictors of sarcopenia, univariate and multivariate logistic regression analyses were conducted. Model 1 was adjusted for traditional risk factors for sarcopenia (age, sex, BMI, and comorbidity index) [16]. Model 2 was adjusted for potential confounders in the association between iron status and sarcopenia (dialysis vintage, CRP levels, Alb levels, Hb levels, comorbidity index, and use of iron supplementation and ESAs) [17,18,19].

All statistical tests were two-sided, and *p*-values of <0.05 were considered significant. All statistical analyses were performed using JMP^®^ 17 (SAS Institute Inc., Cary, NC, USA).

## 3. Results

### 3.1. Patient Characteristics

Among the 104 patients enrolled in this study, 25 (24.0%) were diagnosed with sarcopenia according to the AWGS 2019 criteria. The patient’s baseline characteristics are presented in Table 1. Compared with those without sarcopenia, patients with sarcopenia were older and exhibited a higher frequency of ESA use, elevated serum ferritin levels, a higher comorbidity index, and a lower BMI. No significant differences were observed in other iron status parameters between the two groups.

### 3.2. Serum Ferritin Levels and Sarcopenia

Each 10-ng/mL increment in serum ferritin levels was associated with sarcopenia (unadjusted OR: 1.06, 95% CI: 1.00–1.11, *p* = 0.021); however, other iron status parameters were not associated with sarcopenia (Table 2). The ROC curve analysis indicated that serum ferritin levels were a significant predictor of sarcopenia (area under the ROC curve: 0.651, *p* = 0.019) (Figure 1). In a multivariate model adjusted for traditional risk factors of sarcopenia, a 10 ng/mL increment in serum ferritin levels tended to be associated with sarcopenia but was not deemed significant (OR: 1.05, 95% CI: 0.99–1.12, *p* = 0.071). In the model adjusted for potential confounders related to iron status, the association between a 10 ng/mL increment in serum ferritin levels and sarcopenia remained significant (OR: 1.06, 95% CI: 1.00–1.12, *p* = 0.046).

### 3.3. Association Between High Serum Ferritin Levels and Sarcopenia

The patients were divided into two groups based on an optimal serum ferritin cutoff value of 132 ng/mL, as determined by ROC curve analysis for sarcopenia. The high-ferritin group had a higher prevalence of sarcopenia compared with the low-ferritin group (37.3% vs. 11.3%, *p* = 0.001) (Table 3). Additionally, patients in the high-ferritin group were older, had a lower BMI, and demonstrated a higher frequency of iron supplementation and ESA use compared with those in the low-ferritin group. A serum ferritin level of ≥132 ng/mL was found to be significantly associated with sarcopenia in the logistic regression analyses across multiple models: unadjusted (OR: 4.65, 95% CI: 1.67–12.92, *p* = 0.003), adjusted for traditional risk factors of sarcopenia (OR 5.02, 95% CI 1.47–17.06, *p* = 0.009), and adjusted for potential confounders related to iron status (OR: 5.45, 95% CI: 1.54–19.23, *p* = 0.008) (Table 2).

### 3.4. Association Between Patient Characteristics and Components of Sarcopenia Criteria

Serum ferritin levels were negatively correlated with SMI and handgrip strength (ρ = −0.343, *p* < 0.001 and ρ = −0.253, *p* = 0.009, respectively), whereas no significant correlation was observed between serum ferritin levels and skeletal muscle performance (SPBB) (ρ = −0.148, *p* = 0.13) (Table 4). Age demonstrated significant negative correlations with all three components of sarcopenia (SMI: ρ = −0.432, *p* < 0.001; hand grip: ρ = −0.398, *p* < 0.001; skeletal muscle performance: ρ = −0.236, *p* = 0.015). SMI and handgrip strength were positively correlated with male sex (SMI: ρ = 0.584, *p* < 0.001; hand grip: ρ = 0.534, *p* < 0.001) and BMI (SMI: ρ = 0.622, *p* < 0.001; hand grip: ρ = 0.354, *p* < 0.001). Additionally, the comorbidity index was significantly negatively correlated with physical performance (ρ = −0.204, *p* = 0.037).

## 4. Discussion

This study demonstrated that elevated serum ferritin levels were independently associated with sarcopenia in patients undergoing HD. Although a previous study reported an association between serum ferritin levels and handgrip strength in patients undergoing HD [20], our study applied the AWGS diagnostic criteria, which define sarcopenia as the presence of low muscle mass in combination with reduced muscle strength and/or physical performance. Notably, serum ferritin levels—an indicator not only of iron stores but also of functional iron deficiency—were significantly correlated with skeletal muscle parameters (SMI and handgrip strength). To our knowledge, this study is the first to comprehensively examine and establish an association between serum ferritin levels and sarcopenia based on validated diagnostic criteria in the HD population.

A possible mechanism underlying the association between elevated serum ferritin and sarcopenia may involve iron accumulation-mediated oxidative stress in the skeletal muscle. Ferritin, an iron-storage protein, is translationally regulated by intracellular iron levels. High cellular iron concentrations promote ferritin expression. Therefore, serum ferritin levels serve as an indicator of iron status [2]. Although free cytoplasmic iron is normally maintained at low concentrations by iron-binding proteins such as ferritin, excess free iron is unstable and readily reacts with oxygen and lipid species to generate reactive oxygen species (ROS) [21,22]. The accumulation of ROS is considered a key factor contributing to declines in both muscle quantity and quality [23]. A previous study using human muscle biopsy demonstrated that skeletal muscle iron levels increase with age, promoting ROS generation that contributes to mitochondrial dysfunction and skeletal muscle atrophy [24]. Similarly, a murine model of iron overload showed that iron injections led to increased skeletal muscle iron content and oxidative stress, resulting in decreased muscle strength and muscle atrophy in young mice [25]. These findings and our results suggest that unnecessary elevations in iron levels may contribute to reductions in muscle strength and mass.

Our study demonstrated that patients undergoing HD with serum ferritin levels ≥ 132 ng/mL had 5.02 times higher odds of having sarcopenia after adjusting for age, sex, BMI, and comorbidity index. Previous studies have reported that iron supplementation reduces mortality in this population [26]; hence, excessive iron therapy can result in iron overload [27] and high mortality [28,29]. However, the threshold at which iron overload becomes harmful during iron therapy remains unclear in the HD population. International guidelines for the management of IDA in patients with CKD vary. Guidelines in the United Kingdom and the United States of America recommend withholding iron therapy when serum ferritin levels exceed 500–800 ng/mL [30,31,32,33]. By contrast, Japanese guidelines set a more conservative upper limit of 300 ng/mL for serum ferritin targets [34]. Notably, a previous study involving patients undergoing HD in the United States reported increased risks of all-cause and cardiovascular mortality only at serum ferritin levels exceeding 1200 ng/mL compared with the reference range of 100–199 ng/mL [35]. In contrast, a study on Japanese patients undergoing HD reported an increased risk of infection and cerebrocardiovascular diseases among those with serum ferritin levels > 100 ng/mL [28]. Additionally, data from Japanese patients participating in the Dialysis Outcomes and Practice Patterns Study demonstrated a U-shaped association between serum ferritin levels and mortality, with the 50–99 ng/mL ferritin group showing the most favorable survival outcomes [36]; the study hypothesized that the international differences in iron overload thresholds may be influenced by varying levels of inflammation. Serum ferritin levels in patients undergoing HD are known to increase with inflammation [37], and Japanese patients undergoing HD tend to exhibit lower levels of inflammation compared with those in Western countries, likely due to the higher prevalence of AV fistulas and lower use of catheters and AV grafts [38,39]. The effect of inflammation on ferritin levels may increase the upper limit of serum ferritin considered indicative of iron overload during iron administration in Western countries. A previous study using magnetic resonance imaging to measure liver iron concentration identified a serum ferritin cutoff of 160 ng/mL for iron overload [40], which is lower than the upper limits recommended by international guidelines. Therefore, reassessment of the potential risks associated with high ferritin levels may be necessary, even when serum ferritin levels are below the guideline thresholds. In this study, patients with active infections or cardiovascular diseases requiring hospitalization were excluded. Therefore, the patients in this study may be less influenced by severe inflammation or malnutrition. Inflammation has been demonstrated to cause protein-energy wasting (PEW), the term for loss of body stores of protein and energy fuels [41]. Because serum ferritin levels are thought to have a positive correlation with inflammation, investigation about the relationship between serum ferritin levels and PEW would be important. Further evaluation with the collection of data associated with PEW would be needed in our next study. Our results showed that patients in the high-ferritin group were older, had a lower BMI, and demonstrated higher rates of iron supplementation and ESA use. During ESA treatment, increased iron demand necessitates iron therapy for effective erythropoiesis, which can sometimes contribute to iron overload. Consequently, more patients in the high-ferritin group received iron supplementation and/or ESA compared with those in the low-ferritin group. Particularly in lean elderly patients undergoing ESA therapy, frequent monitoring of iron status and careful adjustment of iron administration strategies may be warranted to prevent adverse events related to iron overload. Given that a rapid increase in serum ferritin levels after the initiation of HD is reportedly associated with higher mortality [42], exploring the association between sarcopenia and changes in serum ferritin levels would be valuable in future research.

This study has some limitations. First, this study was conducted at a single institution in Japan. Hence, the generalizability of the results needs to be verified in other populations. Second, owing to the cross-sectional design, causal relationships cannot be established. Additionally, the lack of data on patients’ activity levels limits the evaluation of the association between inactivity and sarcopenia. However, further prospective studies are needed to validate these findings.

## 5. Conclusions

In conclusion, high serum ferritin levels (≥132 ng/mL) were independently associated with sarcopenia. Notably, this cutoff is lower than the upper limits recommended by international guidelines. This finding underscores the significance of monitoring serum ferritin levels in patients undergoing HD and may suggest that awareness of the potential risks of iron overload and careful evaluation of the need for continuous iron supplementation may be warranted—even when the serum ferritin level remains below the guideline-recommended upper limits. Further prospective intervention studies are necessary to determine the optimal serum ferritin thresholds for initiating and discontinuing iron therapy to prevent sarcopenia and other adverse outcomes in patients undergoing HD with IDA.

## Figures and Tables

**Figure 1 nutrients-17-02323-f001:**
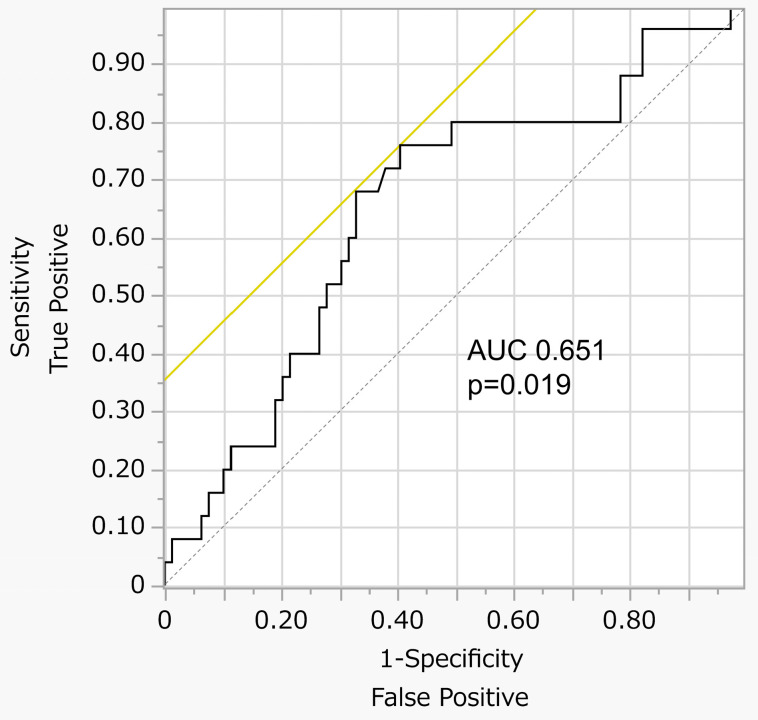
Receiver operating characteristic curve of serum ferritin levels for sarcopenia. Yellow line, the 45-degree tangent line to the curve; Black line, the diagonal reference line.

**Table 1 nutrients-17-02323-t001:** Patient’s baseline characteristics.

	All Patients *n* = 104	Sarcopenia *n* = 25	Non-Sarcopenia *n* = 79	*p*-Value
Age (years)	67.9 ± 11.8	75.2 ± 10.1	64.8 ± 11.3	<0.001
Sex (%), male	78.8	80.0	78.4	0.87
BMI (kg/m^2^)	22.0 ± 3.4	20.4 ± 2.5	22.6 ± 3.5	0.005
Dialysis vintage (years)	7.6 (4.7–14.3)	6.2 (4.1–15.0)	8.2 (5.2–14.3)	0.29
Serum ferritin (ng/mL)	131.3 (59.4–192.3)	170.6 (116.4–222.5)	92 (58.5–182.8)	0.023
Serum iron (µg/dL)	62.7 ± 20.7	61.9 ± 20.9	63.0 ± 20.8	0.79
TIBC (µg/dL)	234 (201.5–260)	237 (211–253.5)	232 (198–260)	0.63
TSAT (%)	27.6 ± 10.6	26.8 ± 10.8	27.8 ± 10.6	0.81
CRP (mg/dL)	0.096 (0.047–0.28)	0.14 (0.063–0.37)	0.083 (0.046–0.28)	0.30
Serum Alb (g/dL)	3.5 ± 0.2	3.5 ± 0.2	3.5 ± 0.2	0.59
Hemoglobin (g/dL)	11.7 ± 1.2	11.4 ± 0.9	11.8 ± 1.2	0.16
Intact PTH (pg/mL)	157 (93.2–225.5)	171 (99–268.5)	155 (81–205)	0.37
ESRD cause				0.28
Diabetes	40.0	52.0	34.1	
Glomerulonephritis	22.0	8.0	25.3	
Hypertensive nephropathy	17.0	16.0	16.4	
PKD	6.0	4.0	6.3	
Other or unknown	19.0	20.0	17.7	
Comorbidity index	4 (2–7)	5 (4–8)	4 (2–7)	0.032
Diabetes (%)	45.1	56.0	41.7	0.21
CVD (%)	29.8	36.0	27.8	0.44
Malignancy (%)	10.6	20.0	7.6	0.098
Liver disease (%)	2.8	8.0	1.3	0.11
Use of iron supplementation (%)	51.9	52.0	51.9	0.99
oral	26.0	24.0	26.6	0.79
intravenous	27.9	28.0	27.9	0.98
Use of ESA (%)	73.1	92.0	67.1	0.007

Data are expressed as the means ± standard deviations or medians (interquartile ranges). BMI, body mass index; TIBC, total iron-binding capacity; TSAT, transferrin saturation; CRP, C-reactive protein; Alb, albumin; PTH, parathyroid hormone; ESRD, end-stage renal disease; PKD, polycystic kidney disease; CVD, cardiovascular disease; ESA, erythropoietin-stimulating agent.

**Table 2 nutrients-17-02323-t002:** Multivariate logistic regression analysis of the association between iron status and sarcopenia.

Variables	Univariate		Model 1 *		Model 2 **	
	OR (95% CI)	*p*-Value	OR (95% CI)	*p*-Value	OR (95% CI)	*p*-Value
Serum ferritin (per 10 ng/mL)	1.06 (1.00–1.11)	0.021	1.05 (0.99–1.12)	0.071	1.06 (1.00–1.12)	0.046
Serum iron (per 10 µg/dL)	0.97 (0.78–1.21)	0.81	1.02 (0.79–1.31)	0.85	1.08 (0.82–1.43)	0.57
TIBC (per 10 µg/dL)	1.03 (0.93–1.13)	0.50	1.11 (0.97–1.26)	0.11	1.05 (0.93–1.19)	0.36
TSAT (%)	0.99 (0.94–1.03)	0.68	0.99 (0.94–1.05)	0.92	1.00 (0.95–1.06)	0.73
Serum ferritin ≥ 132 ng/mL	4.65 (1.67–12.92)	0.003	5.02 (1.47–17.06)	0.009	5.45 (1.54–19.23)	0.008

OR, odds ratio; 95% CI, 95% confidence intervals; TIBC, total iron-binding capacity; TSAT, transferrin saturation. * Model 1 is adjusted for age, sex, body mass index, and comorbidity index. ** Model 2 is adjusted for dialysis vintage, C-reactive protein, albumin, hemoglobin, comorbidity index, and use of iron supplementation and erythropoietin-stimulating agents.

**Table 3 nutrients-17-02323-t003:** Comparison of patient characteristics according to serum ferritin levels.

	Serum Ferritin ≥ 132 ng/mL *n* = 51	Serum Ferritin < 132 ng/mL *n* = 53	*p*-Value
Age (years)	70.3 ± 11.3	64.5 ± 11.8	0.007
Sex (%), male	72.5	84.9	0.12
Sarcopenia (%)	37.3	11.3	0.001
BMI (kg/m^2^)	21.1 ± 2.7	22.9 ± 3.8	0.011
Dialysis vintage (years)	7.5 (4.7–12.9)	7.7 (4.5–14.8)	0.67
CRP (mg/dL)	0.13 (0.050–0.34)	0.079 (0.046–0.26)	0.30
Serum Alb (g/dL)	3.6 ± 0.2	3.5 ± 0.2	0.31
Hemoglobin (g/dL)	11.5 ± 0.8	11.9 ± 1.4	0.099
Intact PTH (pg/mL)	161 (103–230)	154 (66.5–228.5)	0.41
ESRD cause			0.63
Diabetes	35.3	41.5	
Glomerulonephritis	27.4	15.1	
Hypertensive nephropathy	15.7	17.0	
PKD	5.9	5.7	
Other or unknown	15.7	20.7	
Comorbidity index	4 (2–6)	4 (3–7)	0.14
Diabetes (%)	39.2	50.9	0.22
CVD (%)	27.5	32.0	0.60
Malignancy (%)	13.7	7.6	0.30
Liver disease (%)	3.9	1.9	0.53
Use of iron supplementation (%)	62.8	41.5	0.029
oral	33.3	18.9	0.091
intravenous	33.3	22.6	0.22
Use of ESA (%)	88.2	58.5	<0.001

Data are expressed as the means ± standard deviations or medians (interquartile ranges). BMI, body mass index; CRP, C-reactive protein; Alb, albumin; PTH, parathyroid hormone; ESRD, end-stage renal disease; PKD, polycystic kidney disease; CVD, cardiovascular disease; ESA, erythropoietin-stimulating agent.

**Table 4 nutrients-17-02323-t004:** Correlation between variables and components of sarcopenia criteria.

Variables	SMI		Hand Grip		SPBB	
	ρ	*p*-Value	ρ	*p*-Value	ρ	*p*-Value
Age	−0.432	<0.001	−0.398	<0.001	−0.236	0.015
Sex, male	0.584	<0.001	0.534	<0.001	0.054	0.58
BMI	0.622	<0.001	0.354	<0.001	0.0089	0.92
Comorbidity index	0.128	0.19	0.078	0.42	−0.204	0.037
Serum ferritin	−0.343	<0.001	−0.253	0.009	−0.148	0.13
Serum iron	−0.028	0.77	0.050	0.61	0.071	0.47
TIBC	0.160	0.10	0.086	0.38	0.151	0.12
TSAT	−0.110	0.26	−0.0018	0.98	0.012	0.89
CRP	0.063	0.51	−0.070	0.47	−0.156	0.11
Serum Alb	0.164	0.094	−0.022	0.82	0.021	0.82
Hemoglobin	0.070	0.48	0.177	0.071	0.0085	0.93
Intact PTH	−0.084	0.39	−0.156	0.11	0.082	0.40

ρ, Spearman’s correlation coefficient. BMI, body mass index; TIBC, total iron-binding capacity; TSAT, transferrin saturation; CRP, C-reactive protein; Alb, albumin; PTH, parathyroid hormone; SMI, skeletal muscle mass index; SPBB, short physical performance battery.

## Data Availability

To protect the privacy of the participants, the data collected for this study cannot be shared publicly. However, the data are available upon reasonable request from the corresponding author.

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
