# Peer review of "High Serum Ferritin Levels Are Associated with Sarcopenia in Patients Undergoing Chronic Hemodialysis"

_nutrients, 2025, doi:10.3390/nu17142323_

Round 1

Reviewer 1 Report

Comments and Suggestions for Authors
  • The introduction is concise and objective, which is positive. However, it could be slightly expanded to more clearly highlight what is novel about the study and how it contributes to the existing literature.
  • Reading Table 1, it is already evident that the sarcopenia group has higher serum ferritin levels (170 ng/mL) compared to the non-sarcopenia group (92 ng/mL). However, it is not clear what statistical rationale led you to divide patients into two groups based on an optimal serum ferritin cutoff value of 132 ng/mL. This dichotomization should be better justified from a statistical perspective—particularly whether it was truly necessary to support your conclusions. In other words, wouldn’t Tables 1 and 2 alone already be sufficient to conclude that ferritin is relevant to sarcopenia?
  • I assume that you used ROC analysis to identify an optimal cutoff point based on your sample, especially given that there is no clinically established or universally accepted threshold. However, this should be more clearly explained in the manuscript so that readers—particularly those without a statistical background—can understand the rationale and limitations behind this approach. For example, the cutoff of 132 ng/mL was derived from the same dataset used for the outcome analysis, which raises concerns about the generalizability of the results to other populations. Therefore, this grouping should be interpreted as exploratory. Additionally, it would strengthen the manuscript to present the ROC curve and its corresponding statistics (e.g., AUC, sensitivity, specificity, and Youden index) to transparently demonstrate how this cutoff was derived.
  • In the logistic regression analysis, using serum ferritin ≥132 ng/mL as a predictor yielded a more pronounced association (OR close to 4) compared to using serum ferritin (per 10 ng/mL), which showed a more modest OR closer to 1. It would be helpful if the authors could explain this in the manuscript. Was this difference anticipated?
  • Another point is that although the authors reported OR values, they provided limited interpretation of these results. For example, the manuscript does not include a straightforward sentence such as: 'Individuals with ferritin levels above 132 ng/mL had 5.02 times higher odds of having sarcopenia, adjusted for age, sex, body mass index, and comorbidity index.' Including such interpretations would make the findings more accessible.
  • I know that in logistic regression, the OR is derived from the β coefficient of the equation, but this could be more clearly described and detailed in the statistical analysis section.

Reviewer 2 Report

Comments and Suggestions for Authors

Authors present a study, in which they analyzed the correlation between the iron metabolism/ferritin level in hemodialyzed patients with sarcopenia. Authors indicate that ferritin level correlates with sarcopenia in hemodialyzed patients.

Comments:

  1. when showing groups comparison, please if possible show p (statistical significance) values (e.g. L24);
  2. iron deficiency in HD patients is not only related with blood loss though hemodialyzer (L38), please add reference here and add more causes of iron deficiency in HD population;
  3. did you exclude some patients from the study? ferritin is an acute phase protein, what about patients with active infection or malignancy? they may also contribute to sarcopenia;
  4. L66-68 and later: I suggest to use 'level/concentration' in relation to every lab parameter;
  5. Table 1: instead of 'nephrosclerosis' I suggest to use 'hypertensive nephropathy'; please arrange abbreviations in an alphabetical order (also in other Tables and in the list of abbreviations); please change 'GN' into 'glomerulonephritis';
  6. you've mentioned in the discussion about the high ferritin level in correlation with inflammation (L233-237), what about MIA syndrome in some patients?
Comments on the Quality of English Language

Some minor corrections can be done, e.g. L44 I suggest to change 'pathogenesis' into 'development'. 

Round 2

Reviewer 2 Report

Comments and Suggestions for Authors

Thank you very much for improving the manuscript, however mentioning exclusion criteria inyour study in discussing MIA syndrome could potentially increase the clarity of your study.
